# Limited Nitrogen and Plant Growth Stages Discriminate Well Nitrogen Use, Uptake and Utilization Efficiency in Popcorn

**DOI:** 10.3390/plants9070893

**Published:** 2020-07-15

**Authors:** Shahid Khan, Antônio Teixeira do Amaral Júnior, Fernando Rafael Alves Ferreira, Samuel Henrique Kamphorst, Gabriel Moreno Bernardo Gonçalves, Marta Simone Mendonça Freitas, Vanildo Silveira, Gonçalo Apolinário de Souza Filho, José Francisco Teixeira do Amaral, Ricardo Enrique Bresssan Smith, Iftikhar Hussain Khalil, Janieli Maganha Silva Vivas, Yure Pequeno de Souza, Diego Alves Peçanha

**Affiliations:** 1Laboratory of Plant Breeding and Genetics, Center of Agricultural Science and Technology, Darcy Ribeiro State University of Northern Rio de Janeiro, Av. Alberto Lamego, 2000, Campos dos Goytacazes, RJ 28000-000, Brazil; amaraljr@uenf.br (A.T.d.A.J.); samuelhk@pq.uenf.br (S.H.K.); gabriel.agrobio@gmail.com (G.M.B.G.); bressan@uenf.br (R.E.B.S.); janielivivas@gmail.com (J.M.S.V.); yurecrato2007@gmail.com (Y.P.d.S.); 2Department of Plant Breeding and Genetics, University of Agriculture, Peshawar 25130, Pakistan; drihkhalil@aup.edu.pk; 3Laboratory of Phytotechnology, Sector of Mineral and Plant Nutrition, Center of Agricultural Science and Technology, Darcy Ribeiro State University of Northern Rio de Janeiro, Av. Alberto Lamego, 2000, Campos dos Goytacazes, RJ 28013-602, Brazil; fraf.nando@gmail.com (F.R.A.F.); msimone@uenf.br (M.S.M.F.); dialvespecanha@gmail.com (D.A.P.); 4Laboratory of Biotechnology, Center of Biosciences and Biotechnology, Darcy Ribeiro State University of Northern Rio de Janeiro, Av. Alberto Lamego, 2000, Campos dos Goytacazes, RJ 28013-602, Brazil; vanildo@uenf.br (V.S.); goncalos@uenf.br (G.A.d.S.F.); 5Department of Rural Engineering, Center of Agricultural Sciences and Engineering, Federal University of Espírito Santo (UFES), Alegre, ES 29500-000, Brazil; jose.amaral@pq.cnpq.br

**Keywords:** *Zea mays* var. everta, abiotic stress, biomass, sustainability

## Abstract

The extensive use of nitrogen (N) in agriculture has caused negative impacts on the environment and costs. In this context, two pot experiments were performed under different N levels and harvested at different vegetative stages to assess two popcorn inbred lines (P2 and L80) and their hybrid (F1 = P2 × L80) for the N use, uptake and utilization efficiency (with the inclusion and exclusion of root N content); to find the contrasting N levels and vegetative stages that effect nitrogen use efficiency (NUE) and to understand the relationship between the traits related to NUE. The hybrid and P2 were confirmed better than L80 for all the studied traits. NUE is mainly affected by the shoot dry weight, uptake and utilization efficiency. Extremely low and high N levels were found to be more discriminating for N use and dry weight, respectively. At the V6 (six fully expanded leaf) stage, root N content (RNC) should be considered; in contrast, at the VT (tasseling stage) stage, RNC should not be considered for the uptake and utilization efficiency. The genetic parameter performance for N use, uptake, shoot dry weight and N content could favor the achievement of the genetic gain in advanced segregating generations.

## 1. Introduction

Among the varieties belonging to *Zea mays* L., popcorn (*Zea mays* L. var. everta) occupies a prominent position due to containing 583 Kcal/100 g of energy intake and 49% saturated fats, 45% carbohydrates and 7% proteins, while the average price charged by the bag is three times higher than common corn [1,2]. From an agronomic and economic point of view, the primary purpose of crop improvement over the last century has been yield [3]. The enhancing in the crop yield was because of the plant breeding techniques and extensive use of fertilizers. Among these fertilizers, nitrogen (N) is a major factor in agricultural production [4,5].

Nitrogen is important for life on the planet, and it is the most essential nutrient for obtaining high agricultural production. However, nitrogen fertilizers applied in agriculture are not used efficiently by high-yield crops, such as wheat, maize and rice, in which only 33% of the applied N is used by the plant [6]. The use of N leads to the release of nitrous oxide with a global warming potential of 296 times greater than the CO_2_ molecule. The damaged caused by excess use of N in Europe was found to be 91–466 billion US dollars annually, hence, reducing N use in agriculture is a big challenge. Approximately 2.3 billion US dollars could be saved in annual costs of N fertilizer by improving only 1% in N uptake efficiency [7,8].

Demand for the most efficient cultivar in the use of N is desirable and might be possible through the selection of superior genotypes [6,9,10,11]. The idea is to develop N efficient maize cultivars that produce more or the same amount of grains, with less demand for N, and therefore, with lower production costs and environmental impact [12,13]. However, the nutrient use efficiency of popcorn is largely unknown [14]. Exploring the genetic variation for nutrient efficiency should be useful to developed cultivars of popcorn with high N use efficiency (NUE). This can lead to a higher yield, and consequently, reduce the demand for fertilizers [6].

The background on NUE and its components of uptake and utilization based on grain weight was first proposed by Moll et al. [15], who showed that NUE is the ratio of grain weight to N available in the soil or the product of N uptake efficiency (NUpE: the ratio of total N in the plant to N available in the soil) and N utilization efficiency (NUtE: the ratio of grain weight to total N in the plant). Then, Good et al. [16] described, based on the dry weight, that NUE is the ratio of shoot dry weight to N applied, NUpE (ratio of N in the plant to N applied) and NUtE (ratio of shoot dry weight to N in the plant). However, until now, it has been uncertain whether to include the root N content (RNC) or not while measuring the uptake and utilization efficiency, because most of the researchers dealing with these aspects have different conducting methods to include only the shoot N content (N content of leaf and stem) [17,18] or to include the total N content in the plant (N content of shoot and root) [11,15,19].

Previously, Santos et al. [20] evaluated 29 popcorn inbred lines from the germplasm bank of Popcorn Research Breeding Program of Darcy Ribeiro North Fluminense State University (UENF) and performed a field experiment for two different N level at two different locations. In the experiment, based on grain yield, 12, 12 and 5 inbred lines were found to be efficient and responsive, inefficient and non-responsive and intermediate efficient and responsive for N use, respectively. In the current study, two of the most contrasting popcorn inbred lines and their F_1_ hybrid were evaluated under two different pot experiment and harvested at two different plant vegetative stages, with the following objectives: (i) To evaluate the performance of these genotypes for root, shoot dry weight and N content. (ii) To provide the most reliable and specific information about the N use, uptake and utilization efficiency (with the inclusion and exclusion of root N content) based on the dry weight. (iii) To find the contrasting N levels and vegetative stage that effect NUE for further comparative proteomics and molecular study. (iv) To understand the relationship between the traits related to NUE. (v) To evaluate the perspective of genetic gains for NUE by the estimate of genetic parameters.

## 2. Results

### 2.1. Expt. 1: Assessment of Dry Weights and N Content at the V6 Stage

The shoot dry weight (SDW), root dry weight (RDW) and total dry weight (TDW) showed significant (*P* = 0.01) variation among the popcorn genotypes (parents and hybrid). Likewise, the N levels also had a significant (*P* = 0.01) impact on the SDW and TDW except for the RDW. However, the G × N interaction did not influence the SDW, RDW and TDW (Table 1). The lowest SDW and TDW were found in N10% and the highest in N75%, from N75% to N100%, slightly decreased was observed in the SDW and TDW. From N10% to N100%, P2 exhibited an increase of 76.68% and 58.33% for the SDW and TWD, respectively. Hybrid revealed an increase of 29.35% and 18.67% for the SDW and TDW, in contrast, L80 showed an insignificant increase from N10% to N100% (Table 2).

The shoot nitrogen content (SNC), root nitrogen content (RNC) and total nitrogen content (TNC) were significantly different (*P* = 0.01) for genotypes and N levels while being non-significant for G × N interaction (Table 1). At N75%, the SNC, RNC and TNC exhibited high differences for the hybrid and inbred lines P2 and L80. However, at N100%, the hybrid was statistically at par with the inbred line P2. The lowest SNC, RNC and TNC were observed at N10%, while the highest SNC, RNC and TNC were noted at N75%. However, from N75% to N100%, a slight decrease in the SNC, RNC and TNC were observed. From N10% to N100%, P2 exhibited an increase of 140.10% and 126.00%, while L80 demonstrated 192.10% and 171.10% increase of SNC and TNC, respectively. Hybrid revealed an increase of 86.20%, 13.50% and 75.76% for SNC, RNC and TNC, respectively, from N10% to N100% (Table 2).

At the V6 stage, it was observed that the leaf dry weight and leaf N content were greater than stem and root dry weight and N content. With the increasing supply of nitrogen from N10% to N100% the leaf, stem dry weight and N content were augmented more than the root dry weight and N content (Figure 1A,B). This leaf, stem dry weight and N content differences and augmentation were observed more for the inbred line P2 and hybrid than the inbred line L80.

### 2.2. Expt. 2: Assessment of Dry Weights and N Content at the VT Stage

At the VT stage, the shoot dry weight (SDW), root dry weight (RDW) and total dry weight (TDW) showed significant (*P* = 0.01) variation for genotypes, and the N levels significantly (*P* = 0.05) impact only the SDW. The G × N interaction significantly (*P* = 0.05) influenced the RDW and total TDW (Table 1). Hybrid showed a high SDW, RDW and TDW compared to of both their parents (P2 and L80) in N50%. In N100%, the hybrid demonstrated the same performance as P2 and different from L80 for the SDW and TDW, in contrast, for the RDW, these differences were insignificant among the genotypes means in N100%. From N50% to N100%, P2 exhibited an increase of 24.33% and 27.22% for the SDW and TDW, respectively, while the hybrid and L80 showed an insignificant increase (Table 3).

The shoot nitrogen content (SNC) and total nitrogen content (TNC) at the VT stage demonstrated significant (*P* = 0.01) differences among the genotypes and N levels and non-significant for G × N interaction. In contrast, for the root nitrogen content (RNC), the significant (*P* = 0.05) differences were only found for the N levels (Table 1). It was detected from the mean values that under N50%, the SNC and TNC were found more for the hybrid than parents (P2 and L80); in contrast, in N100%, the hybrid was statistically similar to P2 and different from L80. The mean of hybrid and P2 for RNC were found to be different, and the L80 showed similarities with P2 and hybrid in N50%. In N100%, the variations between the mean of P2, L80 and hybrid were found to be alike for the RNC (Table 3). From N50% to N100%, P2 exhibited an increase of 60.53%, 35.32% and 37.56% for the RNC, SNC and TNC, respectively. From N50% to N100%, hybrid revealed an increase of 19.16% and 16.60% for the SNC and TNC, and a 3.80% decrease for the RNC, respectively. In contrast, L80 showed an insignificant increase for the RNC, SNC and TNC from N50% to N100% (Table 3).

At the VT stage, the stem dry weight was found to be higher than the leaf and root dry weight, while the leaf content was observed greater than the stem and root N content (Figure 1C,D); this means that until the VT stage, the leaf still plays a role in the process of photosynthesis. With the increasing supply of nitrogen form N50% to N100%, the leaf, stem dry weight and N content were augmented more than the root dry weight and N content. These leaf, stem dry weight and N content differences and augmentations were observed more for P2 and hybrid than L80, which was also found the same at the V6 stage.

### 2.3. Expt. 1: Reflection of Nitrogen Use, Uptake and Utilization Efficiency at the V6 Leaf Stage

It was quite interesting that nitrogen use efficiency (NUE), nitrogen uptake efficiency with root N content (NUpE-wR) and nitrogen uptake efficiency without root N content (NUpE-w/oR) demonstrated significant (*P* = 0.01) variation among the genotypes, N levels and G × N interaction. When the genotypes were assessed under different N levels, significant (*P* = 0.01) differences were found for the NUE, NUpE-wR and NUpE-w/oR in all the N levels except in N100% (Table 4). Genotypes revealed the highest mean values for the NUE, NUpE-wR and NUpE-w/oR in N10% while the lowest means values were found in N100% (Table 5). From N10% to N100%, P2 exhibited 82.3%, 77.3% and 76.1% of decrease, and L80 showed 79.20%, 72.90% and 70.60% of decrease for the NUE, NUpE-wR and NUpE-w/oR, respectively. Hybrid revealed a decrease of 87.10%, 82.30% and 81.50% from N10% to N100% for the NUE, NUpE-wR and NUpE-w/oR, respectively (Table 5).

Significant (*P* = 0.01) variation was observed for the nitrogen utilization efficiency with root N content (NUtE-wR) among the genotypes and N levels, and similarities were found for the nitrogen utilization efficiency without root N content (NUtE-w/oR) among the genotypes (Table 4). The G × N interaction was found to be non-significant for the NUtE-wR and NUtE-w/oR. The genotypes revealed the highest mean values for the NUtE-wR and NUtE-w/oR in N10%, and the lowest in N100%. From N10% to N100%, P2, L80 and hybrid demonstrated a decrease of 22.80%, 23.40% and 26.60% for the NUtE-wR, respectively (Table 5).

### 2.4. Expt. 2: Reflection of Nitrogen Use, Uptake and Utilization Efficiency at the VT Stage

Significant (*P* = 0.01) dissimilarities were found among the genotypes and N levels for the NUE, NUpE-w/oR and NUpE-wR. The G × N interaction demonstrated significant (*P* = 0.05) differences for the NUE and insignificant for the NUpE-w/oR and NUpE-wR. When the genotypes were studied under N50% and N100% levels, significant (*P* = 0.01) differences were found for the NUE (Table 4). The mean of the hybrid was found statistically superior to both of their parents, and the inbred line P2 was found greater than the inbred line L80 in N50% for the NUE, NUpE-w/oR and NUpE-wR. In contrast, in N100% the hybrid was found to be statistically similar to P2 and different from L80. From N50% to N100%, the inbred line P2 exhibited a reduction of 37.80%, 31.60% and 31.80%, for the NUE, NUpE-w/oR and NUpE-wR, respectively. From N50% to N100%, the inbred line L80 demonstrated a decrease of 46.49%, 42.55% and 42.59% for the NUE, NUpE-w/oR and NUpE-wR, respectively, and the hybrid revealed a decrease of 45.20%, 35.80% and 36.40% for the NUE, NUpE-w/oR and NUpE-wR, respectively (Table 3).

The nitrogen utilization efficiency without root N content (NUtE-w/oR) showed significant (*P* = 0.01) dissimilarities among the genotypes, N levels and G × N interaction. The nitrogen utilization efficiency with root N content (NUtE-wR) exhibited significant (*P* = 0.05) variations among N levels and G × N interaction while similarities were observed among the genotypes. When the genotypes were evaluated under N50% and N100% levels significant (*P* = 0.01) differences were found for the NUtE-w/oR only in N100%, and the NUtE-wR exhibited resemblance in both N levels (Table 4). The mean of the inbred line P2 was statistically found the same as L80 and hybrid but the hybrid and L80 were found different in N100%; in contrast, in N50%, the differences among the genotypes means were found similar for the NUtE-w/oR. From N50% to N100%, P2 exhibited a reduction of 8.40% and 9.80% for the NUtE-w/oR and NUtE-wR, respectively. The inbred line L80 demonstrated a decrease of 5.60% and 6.90% from N50% to N100% for the NUtE-w/oR and NUtE-wR, respectively. The hybrid revealed a decrease of 15.40% and 13.70% from N50% to N100% for the NUtE-w/oR and NUtE-wR, respectively (Table 3).

### 2.5. Genetic Parameters and Principal Component Analysis at the V6 and VT Stages

The coefficients of genetic variation (CVg) vary between 7% and 113% in Expt. 1, in the variables NUtEw/oR and NUE, and between 3% and 30% in Expt. 2, in the variables NUtEw/oR and NUpEw/oR, respectively. In general, CVg, variation index (VI) and H^2^ values were high, except for NUtE-wR and NUtE-w/oR in Expt. 1. The NUtE-wR and RDW, represented genotypic variance component value with a negative sign in Expt. 2, the negative value being replaced by zero, as proposed by Hallauer et al. [21] and Khuri et al. [22]. Additionally, in Expt. 2, the RNC showed higher CVe and low H^2^ of 36.63% and 0.50, respectively, compared to other traits (Table 6).

The principal component (PCA) of Expt. 1 and Expt. 2 of the biplot explained 97.3% and 92.65% of the total variation among the popcorn genotypes for the studied traits across all the N levels, respectively (Figure 2A,B). The length of vectors in Figure 2A,B are alike, which displays that all the traits have a similar magnitude of variation. In both PCAs, the vectors of the shoot, root, total dry weight and N content were found to be closed because of their close association. In contrast, the vectors of nitrogen use, uptake and utilization efficiency were found to be closed because of their close relationship. The head of each vector represented the direction of the highest performance of traits for the genotype in each N level in both of the experiments. Thus, in Expt. 1, the NUE, NUpE-wR, NUpE-woR, NUtE-wR and NUtE-woR were the highest performing traits for inbred line P2 and hybrid in N10%; the SDW, RDW, TDW, SNC, RNC and TNC were the highest performing traits for inbred line P2 in N50%, and hybrid in N50% and N75%. In Expt. 2, NUE, NUpE-wR, NUpE-woR and RDW were the highest performing traits for hybrid in N50%; SDW, SNC, RNC and TNC were the highest performing traits for inbred line P2 and hybrid in N100%. The ellipse in Figure 2A,B showing the same group of treatment belong to the same genotype.

## 3. Discussions

One of the primary concerns of plant breeders and agronomists is to investigate the effectiveness of plants under biotic and abiotic stress conditions [23]. Thus, for this purpose, popcorn genotypes were grown under different N levels and harvested at different vegetative stages. We found, that at both of the stages (V6 and VT), the F_1_ hybrid and inbred line P2 were performing better than inbred line L80 for the SDW, RDW and TDW (Table 2 and Table 3). It is interesting to mention that at both of the stages L80 showed more percentage of change for SNC and TNC from N10% to N100% compared to P2 and hybrid but insignificant change for the SDW, RDW and TDW, which is clear evidence that L80 does not utilize the available N well, and comparatively demonstrated less and insignificant changes for the shoot, root and total dry weight (Table 2 and Table 3). The N content in the shoot or root directly impacts the uptake and inversely impacts the utilization efficiency [17,18]. From Table 1, it was detected that with the increasing N supply, the SNC and RNC increased but the NUpE-wR and NUpE-w/oR decreased instead of increased (Table 5) because the amount of N content found in shoot and root was much less than total N supply or in other words; there was a big gap between the N absorption and supply, which was also evidenced by Ciampitti et al. [24].

It is also noteworthy that at the VT stage the hybrid showed 39.70% and 10.50% of the decrease in the RDW and TDW from N50% to N100%, respectively, which was also documented by Torres et al. [25]. This decrease may be because of heterotic effect and also in low N (N50%) supply, the plants can increase root depth, cortical cell size [26] and root cortical aerenchyma [27,28], which ultimately affect the RDW. It is noteworthy that under N100% the means of L80, P2 and hybrid are alike for the RDW and RNC but not for the SNC and SDW. Hence, it evidenced that at the VT stage, P2 and hybrid translocate and utilized N efficiently than L80 (Table 3).

At the V6 stage, the hybrid showed high performance than P2 and L80, and P2 was found better than L80 for the NUE, NUpE-wR and NUpE-w/oR. This performance was perceived clearly under low N supply of N10% and N25% than high N supply of N75% and N100% (Figure 3), which is because of the increasing N level [29]. At a high concentration of N, all the genotype use and absorb N mostly the same, as evidenced by Mundim et al. [11]. With an increasing supply of N application, the mean values of the NUE, NUpE-wR and NUpE-w/oR were decreased for the genotype as follows N10% > N25% > N50% > N75% > N100%. This reduction was associated with the ratio of SDW to N content in the plant and shoot, respectively, and because of the 25% increase of N applied in each level (Table 2). Fu et al. [30] also stated a 29% decrease for the NUE while studying the effect of five different N rate on sweet-waxy maize in a greenhouse experiment.

The significant differences among the genotypes for the NUtE-wR and non-significant for NUtE-w/oR at the V6 stage means (Table 4) that it is critical to include the root N content (N in the plant) while measuring the utilization efficiency because at the early stages of the plant; the root is also a part of the N content in plants. Additionally, the shoot N converted to biomass was equally utilized by the genotypes in the provided N levels [18]. Kant et al. [8] reported that at the early vegetative stage, leaves and roots operate like a sink for the N uptake and utilization. The product of N uptake and utilization efficiency has a direct measure of the NUE [11,15,16]; however, at the same time, if we are considering NUpE-w/oR and NUtE-w/oR [19], we are ignoring the N content of root, which is also a part of the plant. In the case of uptake or absorption, means the N content that is absorbed by the plant from the available or applied N in the soil or solution. In N utilization the plant utilizes the absorbed N for the root, stem, leaf, growth and grain production; thus, it is noteworthy that at the early stage (V6), measuring NUpE-wR and NUtE-wR [16,19] is a more dependable way of the uptake and utilization efficiency.

It is also worth mentioning that, at the V6 stage, the G × N interaction was found significant for both types of the N uptake efficiency (NUpE-wR and NUpE-w/oR); in contrast, at the VT stage, this G × N interaction was found insignificant (Table 4), which demonstrated that at early stages of the plant (V6), the genotypes were more influenced by different N levels. At the early stages (up to V8), maize plants absorb more N than later stages, which is why fertilization mostly applied up to V6 and V8 stage of the plant [18].

Between the uptake and utilization indices, the results of the genetic parameters revealed a greater genetic variation in the NUpE-wR and NUpE-w/oR measures compared to NUtE-wR and NUtE-w/oR for the two inbred lines. It is suggested that although there is a great contrast between the inbred lines in terms of performance at different N levels, allowing very high values of CVg, VI and H^2^, the indices of utilization are less efficient to reveal these differences. On the other hands, the results demonstrate that there is a strong genetic effect on the traits of dry weight and N content, in all parts of the plants, and on the NUE, NUpE-wR and NUpE-w/oR indices (Table 6).

From the principal component (PCA) Figure 2A,B of the biplot, it was observed the traits related to nitrogen use, uptake and utilization efficiency were well discriminated in low N levels. Thus, the genotypes could be evaluated in low N level for finding high differences for the nitrogen use, uptake and utilization efficiency. In contrast, the dry weight and N content were well distinguished in high N levels. Hence, the genotypes could be assessed in high N level for obtaining high differences of dry weight and N content. Granato et al. [31] reported that a low level of nitrogen and phosphorous provided more chances for genotype discrimination. Hence, genotype evaluations should be performed in low nutrient availability that will lead to a better assessment of genetic variability for nitrogen and phosphorous use. In contrast, the mean of shoot dry weight was decreased by 51% under low N compared to high N. Therefore, High N level was found to be more reliable for discriminating shoot dry weight. Adu et al. [32] documented that low N environment could be regarded as the most discriminating environment for selecting low and high N efficient genotypes.

## 4. Materials and Methods

### 4.1. Experimental Material

The experiments were conducted in the greenhouse (Latitude = 21°9′23″ S; Longitude = 41°10′40″ W; Altitude = 14 m) of Darcy Ribeiro North Fluminense State University (UENF), located in Campos dos Goytacazes, Rio de Janeiro, Brazil, comprising of two popcorn inbred lines P2 and L80 and their F_1_ hybrid (P2 × L80). The plants were evaluated in two growing seasons under different nitrogen levels and harvested at two different stages [20].

### 4.2. Experiment 1

Expt. 1 was performed in January 2018 (first growth season from October to January, with a variable temperature between 25 °C and 38 °C and the relative air humidity ranging from 70 to 76%), using the Hoagland and Arnon [33] solution modified for the N source by nitrate, with five doses: N100% (224.09 mg L^−1^), N75% (168.09 mg L^−1^), N50% (112.04 mg L^−1^), N25% (56.04 mg L^−1^) and N10% (22.40 mg L^−1^) (Table 1). The experiment was conducted in a randomized complete block design with two factorial treatment arrangement (three genotypes × five nitrogen levels) with four blocks. There were three plants per block and one plant per pot in the greenhouse plastic pots (35 L) containing 50% sand washed with deionized water and 50% of fine vermiculite [11,25].

The plants were daily irrigated with deionized water and at V2 (two fully expanded leaves) stage nutrients were provided after every two days according to the plant’s needs [11,12,13,14,15,16,17,18,19]. The plants were harvested at the V6 stage (six fully expanded leaves), and roots, leaves and stem were separated from each plant, which was wrapped in paper bags and dried in the oven at 72 °C for 72 h. Shoot dry weight (SDW, g), root dry weight (RDW, g) and total dry weight (TDW, g) were measured using a high precision digital balance. N concentration was determined by the Kjeldahl method [34] to obtained shoot N content (SNC: SDW × N concentration in the shoot, mg), root N content (RNC: RDW × N concentration in the root, mg) and total N content (TNC: TDW × N concentration in shoot and root or sum of SNC and RNC, mg).

The N use efficiency and its components were calculated as follows; N use efficiency (NUE: SDW/total N applied), N uptake efficiency with root N content (NUpE-wR: N content in the plant (shoot and root)/total N applied), N uptake efficiency without root N content (NUpE-w/oR: shoot N content/total N applied), N utilization efficiency with root N content (NUtE-wR: SDW/N content in the plant) and N utilization efficiency without root N content (NUtE-w/oR: SDW/shoot N content) [11,16,18,19,30,31,32,33,34,35] (Table 7).

### 4.3. Experiment 2

The Expt. 2 was conducted in August 2018 (second growth season from May to August, with a variable temperature between 21 °C and 25 °C and air humidity from 79% to 85%), with the same experimental material using two N doses; N100% (224.09 mg L^−1^) and N50% (112.04 mg L^−1^) (Table 1) and 10 blocks with one plant per block and per pot. The plants were harvested at VT (tasseling) stage and root, leaf and stem were separately wrapped in paper bags and dried in the oven at 72 °C for 72 h. After drying, dry weights, N content, N use efficiency and its components were measured the same as described for Expt. 1.

### 4.4. Statistical Analysis

The data recorded for different traits were subjected to analysis of variance (ANOVA) technique to interpret the main effect (genotypes and N levels) and their interaction, using the following model:**Y_*ijk*l_** = ***μ*** + ***b*_*k*_** + ***g*_*i*_** + ***n*_*j*_** + (***g*** × ***n***)_***ij***_ + ***e*_*ijkl*_**
where **Y_*ijk*l_** = measured phenotype for a given trait of the l^th^ plant of the i^th^ genotype, submitted to the j^th^ nitrogen level, in the k^th^ block, ***μ*** = general constant or grand mean (intercept model); ***b*_*k*_** = random effect of the k^th^ block, where ***b*_*k*_**
*NID* (0, *σb*^2^); ***g*_*i*_** = main effect of the i^th^ genotype; ***n*_*j*_** = the main effect of the j^th^ nitrogen level; (***g*** × ***n***)_***ij***_ = the effects of the interaction between the i^th^ genotype with the j^th^ nitrogen level; and ***e*_*ijkl*_**: = random residual effect, which corresponds the effect of the l^th^ plant nested in the ijk^th^ plot, where ***e*_*ijkl*_**
*NID* (0, *σe*^2^).

Tukey’s test and regression analysis were used for genotype mean comparisons and N levels, respectively (compared to the LSD test, Tukey’s test is more robust and precise and estimates the variance from the whole set of data as a pooled estimate; in contrast, the LSD test estimates the variances only from two of all the groups like individual *t*-tests).

For the estimation of the genetic parameters, analysis of variance was performed including only the inbred lines P2 and L80 and nitrogen levels, in both experiments. From the ANOVA mean squares between the inbred lines and N levels, the following parameters were estimated: the phenotypic variance component, with the estimator σ^2^P = MSg/r (where MSg is the genotype mean square and r is the number of repetitions); genotypic variance component, with the estimator σ^2^G = (MSg − MSe)/r (where MSe corresponds to the error mean square); coefficient of experimental variation, with the estimator CVe% = 100.(MSe)^1/2^/µ (where µ is the average of treatments); coefficient of genotypic variation, with the estimator CVg% = 100.(σ^2^G)^1/2^/µ; variation index, with the estimator VI = CVg/CVe; and heritability, with the estimator H² = σ^2^G/σ^2^P.

For a better understanding of the relationship between the variables, graphs of the principal component (PCA) were generated for each environment, including all the traits evaluated. All the analyses were performed with R software [36], and for graphs, Prism 8.0 was used.

## 5. Conclusions

It was concluded that at both stages, the F_1_ hybrid and the inbred line P2 were found to be close and better than the inbred line L80 for N use, uptake, utilization, dry weight and N contents. Measuring NUE based on dry weight is mainly affected by shoot dry weight, uptake and utilization efficiency. In both of the stages, the leaf N content was found more than the root and stem N content, and hybrid mostly demonstrated high performance for all the studied traits compare to their parents. The traits related to nitrogen use, uptake and utilization efficiency were well discriminated in low N levels. Thus, the genotypes could be evaluated in low N level for finding high differences for the nitrogen use, uptake and utilization efficiency. In contrast, the dry weight and N content were well distinguished in high N levels. Hence, the genotypes could be assessed in high N level for obtaining high differences of dry weight and N content. Regarding the N uptake and utilization efficiency, the results further suggest, at the early stage (V6), root N content (RNC) should be considered while measuring the uptake and utilization efficiency, and the plants absorb more N at the V6 than VT stage. At the later stage (VT), RNC should not be considered for uptake and utilization efficiency. The highest values of coefficient of genotypic variation, variation index and heritability for the N use, uptake, shoot dry weight and N content at both of the stages could favor the achievement of the genetic gain in advanced segregating generations.

## Figures and Tables

**Figure 1 plants-09-00893-f001:**
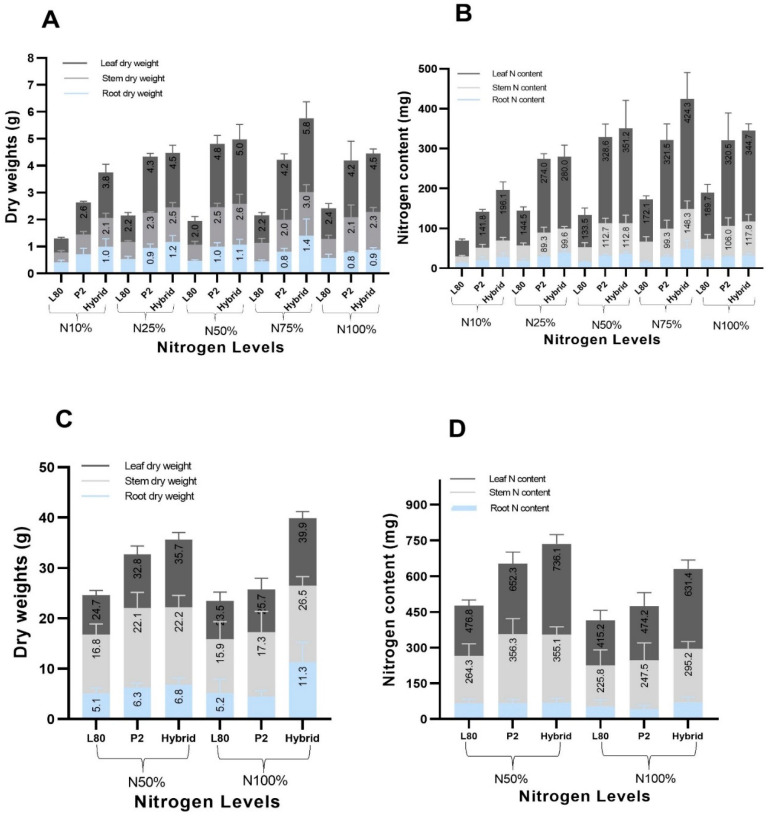
Comparison of the leaf, stem, root dry weight (**A**), N content (**B**) at the V6 stage, leaf and stem, root dry weight (**C**) and N content (**D**) at the VT stage of the popcorn genotypes under different N levels.

**Figure 2 plants-09-00893-f002:**
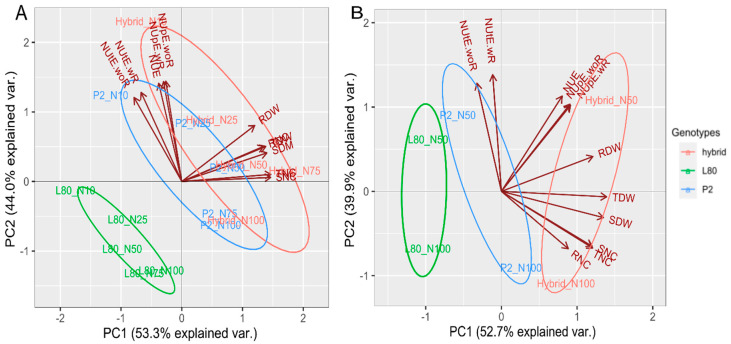
Principal component analysis (PCA) Expt. 1 (**A**) and Expt. 2 (**B**).

**Figure 3 plants-09-00893-f003:**
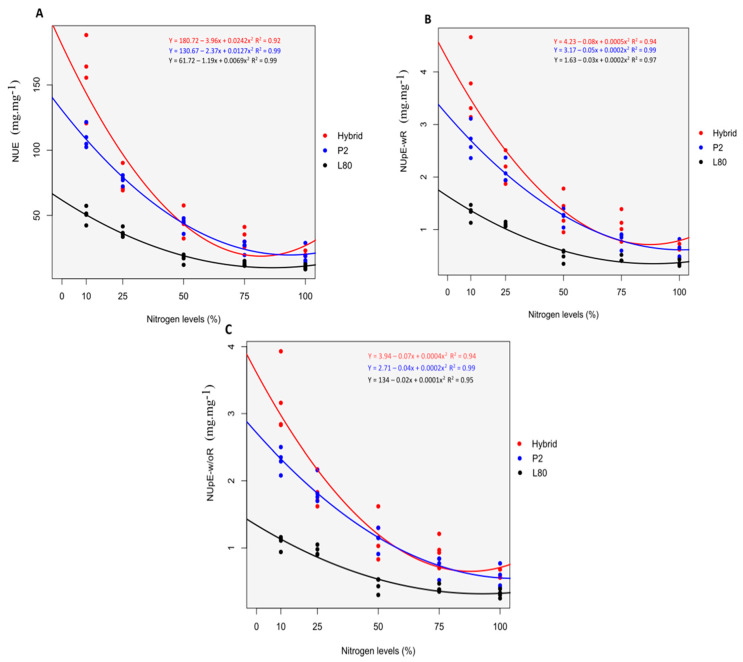
Performance of the popcorn genotypes under five different N levels for NUE (**A**), NUpE-w/oR (**B**) and NUpE-wR (**C**) at the V6 stage.

**Table 1 plants-09-00893-t001:** Analysis of variance of popcorn genotypes (G), nitrogen levels (N) and G × N interaction for the shoot, root, total dry weight and N content at the V6 and VT stage.

Expt. 1, V6 (Six Fully Expanded Leaves) Stage							
Source of Variation	DF	Mean Squares					
		SDW (g)	RDW (g)	TDW (g)	SNC (mg)	RNC (mg)	TNC (mg)
Block	3	0.11	0.14	0.11	2736.43	280.83	1992.72
Genotypes (G)	2	24.23 **	1.91 **	39.32 **	137,857.01 **	1971.02 **	171,674.13 **
Nitrogen (N)	4	3.34 **	0.08 ^ns^	4.14 **	48,291.04 **	192.24 **	54,079.61 **
G × N	8	0.41 ^ns^	0.08 ^ns^	0.75 ^ns^	3458.28 ^ns^	86.15 ^ns^	4295.10 ^ns^
Error	42	0.3	0.043	0.47	2219.34	48.03	2609.62
CV (%)		19.93	25.51	19.28	21.5	25.52	20.75
**Expt. 2, VT (Tasseling Stage) Stage**							
Block	9	22.28	6.78	43.4	10,870.44	412.19	13,947.48
Genotypes (G)	2	482.63 **	96.95 **	963.06 **	259,900.06 **	1111.76 ^ns^	282,576.82 *
Nitrogen (N)	1	76.32 *	12.66 ^ns^	26.81 ^ns^	159,942.51 **	2010.02 *	197,812.42 **
G × N	2	33.35 ^ns^	52.44 **	156.82 **	13,316.21 ^ns^	993.00 ^ns^	17,353.69 ^ns^
G/N	4		74.70 **	559.944 **			
G/N50%	2		141.67 **	792.30 **			
G/N100%	2		7.72 ^ns^	327.58 **			
N/G	3		39.17 **	113.48 *			
N/P2	1		16.65 ^ns^	245.63 **			
N/L80	1		0.04 ^ns^	7.09 ^ns^			
N/Hybrid	1		100.85 **	87.73 ^ns^			
Error	45	17.14	4.28	29.93	7083.81	378.17	9198.11
CV (%)		17.37	31.68	18.02	16.73	31.73	16.99

**, * = Significant at 0.01 and 0.05 probability, respectively, ^ns^ = non-significant. SDW: shoot dry weight, RDW: root dry weight, TDW: total dry weight, SNC: shoot N content, RNC: root N content and TNC: total N content.

**Table 2 plants-09-00893-t002:** Mean values of popcorn genotypes for the shoot, root, total dry weight, and N content under five different N levels at the V6 stage.

					Nitrogen Levels					
Genotypes	N10%		N25%		N50%		N75%		N100%	
**SDW (g)**										
**P2**	1.93	Ba	3.39	Aa	3.81	Aa	3.42	Aa	3.41	Aa
**L80**	0.89	Ab	1.62	Ab	1.49	Ab	1.72	Ab	1.84	Ab
**Hybrid**	2.76	Ba	3.30	ABa	3.92	Aa	4.36	Aa	3.57	ABa
**RDW (g)**										
**P2**	0.72	Aab	0.95	Aa	1.00	Aa	0.80	Ab	0.78	Aa
**L80**	0.41	Ab	0.54	Ab	0.46	Ab	0.45	Ab	0.58	Aa
**Hybrid**	0.99	ABa	1.17	ABa	1.06	ABa	1.40	Aa	0.89	Ba
**TDW (g)**										
**P2**	2.64	Ba	4.33	Aa	4.81	Aa	4.22	Ab	4.18	Aa
**L80**	1.30	Ab	2.16	Ab	1.95	Ab	2.17	Ac	2.42	Ab
**Hybrid**	3.75	Ba	4.48	ABa	4.98	ABa	5.76	Aa	4.45	ABa
**SNC (mg)**										
**P2**	121.41	Bab	244.74	Aa	296.8	Aa	292.62	Ab	291.51	Aa
**L80**	57.32	Bb	126.38	ABb	117.88	ABb	156.4	Ac	167.41	Ab
**Hybrid**	168.00	Ca	241.20	BCa	314.95	ABa	375.83	Aa	312.79	ABa
**RNC (mg)**										
**P2**	20.37	Aab	29.23	Aab	31.77	Aa	28.93	Ab	28.94	Aa
**L80**	12.65	Ab	18.15	Ab	15.64	Ab	15.74	Ac	22.27	Aa
**Hybrid**	28.14	Ba	38.76	ABa	36.27	ABa	48.42	Aa	31.94	Ba
**TNC (mg)**										
**P2**	141.78	Bab	273.97	Aa	328.57	Aa	321.55	Ab	320.45	Aa
**L80**	69.97	Bb	144.53	ABb	133.51	ABb	172.14	ABc	189.68	Ab
**Hybrid**	196.14	Ca	279.96	BCa	351.21	ABa	424.25	Aa	344.73	ABa

Means followed by the same uppercase letter in the row and lowercase letter in the column do not differ statistically by Tukey’s test at 5% probability from one another for N Levels and genotypes, respectively. SDW: shoot dry weight, RDW: root dry weight, TDW: total dry weight, SNC: shoot N content, RNC: root N content and TNC: total N content.

**Table 3 plants-09-00893-t003:** Mean values of popcorn genotypes for the dry weight, N content, nitrogen use, uptake and utilization efficiency under two different N level at the VT stage.

Genotypes	N50%		N100%		N50%		N100%	
**NUE (mg.mg^−1^)**					**NUpE-wR (mg.mg^−1^)**			
**P2**	**29.69**	Ab	18.46	Ba	0.66	Ab	0.45	Ba
**L80**	23.62	Ac	12.64	Bb	0.54	Ac	0.31	Bb
**Hybrid**	39.85	Aa	21.84	Ba	0.88	Aa	0.56	Ba
**NUpE-w/oR (mg.mg^−1^)**					**NUtE-wR (mg.mg^−1^)**			
**P2**	0.60	Ab	0.41	Ba	45.04	Aa	40.61	Ba
**L80**	0.47	Ac	0.27	Bb	44.23	Aa	41.17	Ba
**Hybrid**	0.78	Aa	0.50	Ba	45.41	Aa	39.17	Ba
**NUtE-w/oR (mg.mg^−1^)**					**SDW (g)**			
**P2**	49.48	Aa	45.31	Bab	21.29	Bb	26.47	Aa
**L80**	50.56	Aa	47.75	Ba	18.26	Ab	19.54	Ab
**Hybrid**	51.05	Aa	43.19	Bb	28.58	Aa	28.88	Aa
**RDW (g)**					**TDW (g)**			
**P2**	4.46	Ab	6.28	Aa	25.75	Bb	32.76	Aa
**L80**	5.20	Ab	5.11	Aa	23.46	Ab	24.65	Ab
**Hybrid**	11.32	Aa	6.83	Ba	39.89	Aa	35.70	Aa
**SNC (mg)**					**RNC (mg)**			
**P2**	432.09	Bb	584.73	Aa	42.11	Bb	67.60	Aa
**L80**	361.63	Ab	411.32	Ab	53.57	Aab	65.50	Aa
**Hybrid**	560.55	Ba	668.00	Aa	70.80	Aa	68.11	Aa
**TNC (mg)**					**-**			
**P2**	474.19	Bb	652.33	Aa	-	-	-	-
**L80**	415.21	Ab	476.82	Ab	-	-	-	-
**Hybrid**	631.35	Ba	736.12	Aa	-	-	-	-

Means followed by the same uppercase letter in the row and lowercase letter in the column do not differ statistically by Tukey’s test at 5% probability from one another for N Levels and genotypes, respectively. NUE: nitrogen use efficiency, NUpE-wR: nitrogen uptake efficiency with root N content, NUpE-w/oR: nitrogen uptake efficiency without root N content, NUtE-wR: nitrogen utilization efficiency with root N content, NUtE-w/oR: nitrogen utilization efficiency without root N content SDW: shoot dry weight, RDW: root dry weight, TDW: total dry weight, SNC: shoot N content, RNC: root N content and TNC: total N content.

**Table 4 plants-09-00893-t004:** Analysis of variance of popcorn genotypes (G), nitrogen levels (N) and G × N interaction for nitrogen use, uptake and utilization efficiency at the V6 and VT stage.

Expt. 1, V6 (Six Fully Expanded Leaves) Stage						
	Mean Squares					
Source of Variation		NUE (mg.mg^−1^)	NUpE-wR (mg.mg^−1^)	NUpE-w/oR (mg.mg^−1^)	NUtE-wR (mg.mg^−1^)	NUtE-w/oR (mg.mg^−1^)
Block	3	42.07	0.06	0.05	8.26	16.01
Genotypes (G)	2	8796.16 **	5.71 **	4.47 **	33.94 **	21.13 ^ns^
Nitrogen (N)	4	15,826.00 **	8.34 **	5.83 **	189.39 **	371.46 **
G × N	8	1553.32 **	0.69 **	0.53 **	2.99 ^ns^	2.90 ^ns^
G/N	10	3001.89 **	1.69 **	1.32 **		
G/N10%	2	11,425.00 **	5.78 **	4.45 **		
G/N25%	2	2070.69 **	1.35 **	1.05 **		
G/N50%	2	981.12 **	0.83 **	0.68 **		
G/N75%	2	415.10 *	0.41 **	0.31 **		
G/N100%	2	117.53 ^ns^	0.10 ^ns^	0.09 ^ns^		
N/G	12	6310.88 **	3.24 **	2.30 **		
N/P2	4	5723.31 **	3.09 **	2.22 **		
N/L80	4	1203.49 **	0.77 **	0.50 **		
N/Hybrid	4	12,005.85 **	5.87 **	4.17 **		
Error	42	88.62	0.07	0.04	5.16	6.98
CV (%)		19.24	19.38	17.86	6.62	6.78
**Expt. 2, VT (Tasseling Stage) Stage**						
Block	9	32.37	0.02	0.02	24.72	25.77
Genotypes (G)	2	809.58 **	0.44 **	0.38 **	1.54 ^ns^	24.45 **
Nitrogen (N)	1	2695.68 **	0.96 **	0.75 **	314.42 **	367.09 *
G × N	2	79.54 *	0.02 ^ns^	0.010 ^ns^	12.70 *	34.18 **
G/N	4	444.56 **			7.12 ^ns^	29.32 **
G/N50%	2	672.55 **			3.64 ^ns^	6.41 ^ns^
G/N100%	2	216.57 **			10.60 ^ns^	52.23 **
N/G	3	951.59 **			113.27 **	145.15 **
N/P2	1	630.34 **			98.14 **	87.15 **
N/L80	1	602.80 **			46.92 **	39.34 *
N/Hybrid	1	1621.62 **			194.77 **	308.97 **
Error	45	22.96	0.01	0.01	3.90	6.23
CV (%)		19.68	19.26	19.05	4.63	5.21

**, * = Significant at 0.01 and 0.05 probability, respectively, ^ns^ = non-significant. NUE: nitrogen use efficiency, NUpE-wR: nitrogen uptake efficiency with root N content, NUpE-w/oR: nitrogen uptake efficiency without root N content, NUtE-wR: nitrogen utilization efficiency with root N content and NUtE-w/oR: nitrogen utilization efficiency without root N content.

**Table 5 plants-09-00893-t005:** Mean values of popcorn genotypes for N use efficiency (NUE) and its components of uptake and utilization under five different N levels at the V6 stage.

					Nitrogen Levels					
Genotypes	N10%		N25%		N50%		N75%		N100%	
**NUE (mg.mg^−1^)**										
**P2**	109.66	Ab	77.19	Ba	43.44	Ca	25.98	CDab	19.40	Da
**L80**	50.41	Ac	36.86	Ab	16.93	Bb	13.03	Bb	10.51	Ba
**Hybrid**	157.08	Aa	75.27	Ba	44.63	Ca	33.13	CDa	20.32	Da
**NUpE-wR (mg.mg^−1^)**										
**P2**	2.69	Ab	2.08	Ba	1.25	Ca	0.81	CDab	0.61	Da
**L80**	1.33	Ac	1.10	Ab	0.51	Bb	0.44	Bb	0.36	Ba
**Hybrid**	3.72	Aa	2.13	Ba	1.33	Ca	1.07	CDa	0.66	Da
**NUpE-w/oR (mg.mg^−1^)**										
**P2**	2.30	Ab	1.86	Ba	1.13	Ca	0.74	CDab	0.55	Da
**L80**	1.09	Ac	0.96	Ab	0.45	Bb	0.40	Bb	0.32	Ba
**Hybrid**	3.19	Aa	1.83	Ba	1.20	Ca	0.95	CDa	0.59	Da
**NUtE-wR (mg.mg^−1^)**										
**P2**	40.91	Aab	37.27	ABa	34.80	BCa	31.98	Ca	31.60	Ca
**L80**	38.01	Ab	33.53	ABa	33.54	ABa	29.94	Ba	29.11	Ba
**Hybrid**	42.27	Aa	35.49	Ba	33.77	Ba	31.11	Ba	31.03	Ba
**NUtE-w/oR (mg.mg^−1^)**										
**P2**	47.60	Aa	41.82	Ba	38.63	BCa	35.35	Ca	34.89	Ca
**L80**	46.32	Aa	38.36	Ba	38.20	BCa	32.96	Ca	33.17	BCa
**Hybrid**	49.30	Aa	41.13	Ba	37.80	BCa	35.01	Ca	34.24	Ca

Means followed by the same uppercase letter in the row and lowercase letter in the column do not differ statistically by Tukey’s test at 5% probability from one another for N Levels and genotypes, respectively. NUE: nitrogen use efficiency, NUpE-wR: nitrogen uptake efficiency with root N content, NUpE-w/oR: nitrogen uptake efficiency without root N content, NUtE-wR: nitrogen utilization efficiency with root N content and NUtE-w/oR: nitrogen utilization efficiency without root N content.

**Table 6 plants-09-00893-t006:** Genetic parameters for dry weights, N content, nitrogen use, uptake and utilization efficiency at the V6 and VT stage.

	Expt. 1, V6 Stage (Two Inbreed Lines, Five N Levels and Four Repetitions)										
Parameters	NUE	NUpE-wR	NUpE-w/oR	NUtE-wR	NUtE-w/oR	RDW	SDW	TDW	SNC	RNC	TNC
σ^2^P	2113.25	1.37	1.14	15.49	8.61	0.32	7.07	10.40	38,649.25	300.22	45,762.25
σ^2^G	2106.65	1.37	1.14	14.49	7.36	0.32	7.01	10.34	38,262.50	296.59	45,371.75
CVe	12.75	14.67	13.71	5.87	5.78	15.04	20.49	16.71	21.01	17.04	18.86
CVg	113.78	104.61	108.85	11.17	7.00	84.51	112.57	106.55	104.47	76.99	101.62
VI	8.92	7.13	7.94	1.90	1.21	5.62	5.49	6.38	4.97	4.52	5.39
H^2^	1.00	1.00	1.00	0.94	0.85	0.99	0.99	0.99	0.99	0.99	0.99
	**Expt. 2, VT Stage (Two Inbreed Lines, Two N Levels and Ten Repetitions)**										
σ^2^P	35.28	0.02	0.02	0.02	3.10	0.05	24.77	27.00	14,867.90	13.63	13,747.80
σ^2^G	32.20	0.02	0.02	−0.30	2.48	−0.20	22.59	23.57	13,978.50	6.91	12,607.40
CVe	26.33	25.35	25.43	4.12	5.17	29.69	21.82	21.96	21.08	36.63	21.16
CVg	26.89	26.59	30.65	-	3.26	-	22.22	18.21	26.42	11.75	22.25
VI	1.02	1.05	1.21	-	0.63	-	1.02	0.83	1.25	0.32	1.05
H^2^	0.91	0.93	0.94	-	0.80	-	0.91	0.87	0.94	0.51	0.92

NUE: nitrogen use efficiency, NUpE-wR: nitrogen uptake efficiency with root N content, NUpE-w/oR: nitrogen uptake efficiency without root N content, NUtE-wR: nitrogen utilization efficiency with root N content, NUtE-w/oR: nitrogen utilization efficiency without root N content SDW: shoot dry weight, RDW: root dry weight, TDW: total dry weight, SNC: shoot N content, RNC: root N content, TNC: total N content, σ^2^P: phenotypic variance component, σ^2^G: genotypic variance component, CVe: coefficient of experimental variation, CVg: coefficient of genotypic variation and VI: variation.

**Table 7 plants-09-00893-t007:** Solution with five different contrasting N levels, based on the Hoagland and Arnon Solution, (1950).

Stock Solution	Concentration (mL L^−1^)				
	10%	25%	50%	75%	100%
**Ca(NO_3_)_2_ 4H_2_O (2 mol L^−1^)**	0.4	1	2	2	2
**KNO_3_ (2 mol L^−1^)**	-	-	-	2	2
**MgSO_4_ (1 mol L^−1^)**	2	2	2	2	2
**FeEDTA (25 g L^−1^)**	1	1	1	1	1
**Micro ***	1	1	1	1	1
**H_3_BO_3_ (25 mM)**	1	1	1	1	1
**NaNO_3_ (2 mol L^−1^)**	-	-	-	-	2
**CaCl_2_ (2 mol L^−1^)**	1.6	1	-	-	-
**KCl (1 mol L^−1^)**	4	4	4	-	-
**K_2_HPO_4_ (1 mol L^−1^)**	1	1	1	1	1
**K_2_SO_4_ (0, 5 mol L^−1^)**	1	1	1	1	1

* Micro: CuSO_4_5H_2_O = 125 mg L^−1^; KCl = 3728 mg L^−1^; MnSO_4_H_2_O = 845 mg L^−1^; ZnSO_4_7H_2_O = 578 mg L^−1^; (NH_4_)_6_Mo_7_O_24_4H_2_O = 88 mg L^−1^.

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
