# Peer review of "Limited Nitrogen and Plant Growth Stages Discriminate Well Nitrogen Use, Uptake and Utilization Efficiency in Popcorn"

_plants, 2020, doi:10.3390/plants9070893_

Round 1

Reviewer 1 Report

The revised manuscript entitled “Limited nitrogen and plant growth stages discriminate well nitrogen use, uptake and utilization efficiency in popcorn” investigates the NUE of inbred lines and their hybrids of popcorn. The subject is a very important one as the NUE is important for the efficient use of N as it is one of the main inputs in agriculture. The authors made a significant effort to revised their manuscript and follow the reviewer’s comments. Therefore, I recommend acceptance of the manuscript.

Author Response

Thank you for your feedback on our article. Your previous critical comments contributed a lot.

Reviewer 2 Report

The work is interesting and well explained, and requires only minor revision. Some suggestions follow.

The English is generally good; anyway, some grammar and typing mistakes are present (especially in the discussion section) and should be corrected. Accurate text editing is advisable.

Line 103. Total nitrogen content should be TNC in parentheses.

Line 248 – 249. “…hybrid in N50% and N100%”. From Fig. 2A, the treatments seem to be N50 and N75.

Line 362. What is the "N concentration in shoot and root"?  N concentration in shoot is generally different from N concentration in root; therefore, TNC should be the sum of SNC and RNC.

Line 372. Why where N50% and N100% chosen? Why were lower N doses not included in Expt. 2?

Line 373. How many leaves were present at tasseling stage?

Line 390. Incomplete sentence.

Author Response

Response to Reviewer Comments

First of all, thank you so much for taking the time reviewing our manuscript. I am sure your considerations and corrections will give more reading comprehension and add more value to the paper. On behalf of the other authors, I appreciate the feedback from the Reviewer.

All suggestions made by the Reviewer were taken into account and the responses are highlighted with yellow color.

The English is generally good; anyway, some grammar and typing mistakes are present (especially in the discussion section) and should be corrected. Accurate text editing is advisable.

Response 1. The whole manuscript was revised for typing and English grammar mistakes.

Line 103. Total nitrogen content should be TNC in parentheses.

Response 2. Thanks for the comment and RNC was replaced by TNC.

Line 248 – 249. “…hybrid in N50% and N100%”. From Fig. 2A, the treatments seem to be N50 and N75.

Response 3. Thanks, we agree with the reviewer and the treatment N100% was replaced by N75%.

Line 362. What is the "N concentration in shoot and root"?  N concentration in shoot is generally different from N concentration in root; therefore, TNC should be the sum of SNC and RNC.

Response 4. Well, N concentration multiplied by its respective dry weight is known as N content. That’s why we calculated the total N content by this way TNC: TDW × N concentration in shoot and root. But we agree the TNC should also be the sum of SNC and RNC and we already add this in the methodology.

Line 372. Why were N50% and N100% chosen? Why were lower N doses not included in Expt. 2?

Response 5. Thanks for the comments. Well, we do not include the lower N doses in Exp. 2 because we harvested the plants at tasseling stage and lower dose like N10% gives much stress to the L80 inbred line so it was a risk to lose the plant or leaves to reach the tasseling stage and also at the high-stress condition it’s difficult to harvest all the genotype at the same leaf stage. Moreover, the experiments were performed in the greenhouse with deionized water so keeping these conditions we decided to do not use the lower doses at tasseling stage.

Line 373. How many leaves were present at tasseling stage?

Response 6. At the tasseling stage thirteen, fully expanded leaves were present (excluding the upper two non-collar leaves)

Line 390. Incomplete sentence.

Response 7. The sentence was rewritten.
